# *Nr1d1* Mediated Cell Senescence in Mouse Heart-Derived Sca-1^+^CD31^−^ Cells

**DOI:** 10.3390/ijms232012455

**Published:** 2022-10-18

**Authors:** Shiming Pu, Qian Wang, Qin Liu, Hongxia Zhao, Zuping Zhou, Qiong Wu

**Affiliations:** 1Guangxi Universities Key Laboratory of Stem Cell and Biopharmaceutical Technology, Guangxi Normal University, Guilin 541004, China; 2Research Center for Biomedical Sciences, Guangxi Normal University, Guilin 541004, China; 3Key Laboratory of Ecology of Rare and Endangered Species and Environmental Protection, Guangxi Normal University, Ministry of Education, Guilin 541004, China; 4School of Life Sciences, Guangxi Normal University, Guilin 541004, China; 5Faculty of Biological and Environmental Sciences, University of Helsinki, 00790 Helsinki, Finland

**Keywords:** Sca-1^+^CD31^−^ cells, *Nr1d1*, cell senescence, *Serpina3*, *Nr4a3*

## Abstract

Aim: Sca-1^+^CD31^−^ cells are resident cardiac progenitor cells, found in many mammalian tissues including the heart, and able to differentiate into cardiomyocytes in vitro and in vivo. Our previous work indicated that heart-derived Sca-1^+^CD31^−^ cells increased the Nr1d1 mRNA level of *Nr1d1* with aging. However, how *Nr1d1* affects the senescence of Sca-1^+^CD31^−^ cells. Methods: Overexpression and knockdown of *Nr1d1* in Sca-1^+^CD31^−^ cells and mouse cardiac myocyte (MCM) cell lines were performed by lentiviral transduction. The effects of *Nr1d1* abundance on cell differentiation, proliferation, apoptosis, cell cycle, and transcriptomics were evaluated. Moreover, binding of *Nr1d1* to the promoter region of *Nr4a3* and *Serpina3* was examined by a luciferase reporter assay. Results and Conclusions: Upregulation *Nr1d1* in young Sca-1^+^CD31^−^ cells inhibited cell proliferation and promoted apoptosis. However, depletion of *Nr1d1* in aged Sca-1^+^CD31^−^ cells promoted cell proliferation and inhibited apoptosis. Furthermore, *Nr1d1* was negatively associated with cell proliferation, promoting apoptosis and senescence-associated beta-galactosidase production in MCMs. Our findings show that *Nr1d1* stimulates *Serpina3* expression through its interaction with *Nr4a3*. *Nr1d1* may therefore act as a potent anti-aging receptor that can be a therapeutic target for aging-related diseases.

## 1. Introduction

NR1D1 (also known as *Rev-erbα*) encodes a nuclear heme receptor, belonging to the nuclear receptor superfamily of ligand responsive transcription factors [1], and is involved in circadian rhythm, metabolic, inflammatory, and cardiovascular processes [2,3]. NR1D1 exerts its control via interaction with DNA targets as a potent repressor of transcription [4]. Many researchers have found a wide variety of *Nr1d1*-regulated genes implicated in key cellular processes. Rev-erb, for example, directly suppresses the expression of the positive clock components Bmal1 and Clock [5,6] as well as *Ucp1*, *ApoA1*, and *ApoCIII*, which regulate body temperature cycles and cholesterol metabolism [4,7]; and PARP1, which is responsible for DNA repair of ROS-induced DNA damage in cancer cells [8]. Overexpression of Rev-erbα in C2C12 was shown to increase mitochondrial content and activity, responding to changes in energy availability [9].

The Sca-1^+^CD31^−^ cells are shown to be resident cardiac progenitor cells, found in many mammalian tissues including the heart, and is able to differentiate into cardiomyocytes in vitro [10,11] and in vivo [12,13,14]. Sca-1^+^CD31^−^ cells experience age-related changes, including decreased differentiation ability, proliferation capacity, and increased apoptosis. We have previously proved that the mRNA abundance of *Nr1d1* increased with age, particularly in aged heart-derived Sca-1^+^CD31^−^ cells [14]. The agonist of *Nr1d1* is considered to be an effective anti-tumor and anti-inflammatory strategy [15,16]. However, the high expression of *Nr1d1* in the hearts of elderly mice suggests that Nr1d1 plays an important role in heart aging. Currently, little is known about how NR1D1 contributes to the heart-derived Sca-1^+^CD31^−^ cell plasticity during aging, and the regulatory mechanisms of *Nr1d1* remain to be revealed.

## 2. Results

### 2.1. Nr1d1 mRNA Expression during Aging

Through the analysis of two independent datasets from the GEO database (GSE43556 and GSE7196), we found that the *Nr1d1* mRNA level was significantly upregulated in the aged mouse heart tissue compared to the young tissue (*p* = 0.0011). To investigate the role of *Nr1d1* in aging, we used qRT-PCR to detect *Nr1d1* expression in the heart, liver, bone marrow, brain, and adipose tissues of different aged mice. The results showed a significantly increased expression of *Nr1d1*, especially in aged heart tissue when compared to heart tissue of young mice. Changes in *Nr1d1* abundance in other tissue samples were not statistically significant across ages (Figure 1A). Our previous mRNA expression profiles of heart derived Sca-1^+^CD31^−^ cells at 2 and 20 months showed that the mRNA level of *Nr1d1* increased significantly (7.96 fold) in aged heart derived Sca-1^+^CD31 cells [14]. Notably, *Nr1d1* was expressed predominantly in the aged heart-derived Sca-1^+^CD31 cells.

### 2.2. Nr1d1 Was Expressed Predominantly in the Aged Heart-Derived Sca-1^+^CD31^−^ Cells

In order to assess the function of NR1D1, we performed gene knockdown and overexpression experiments using transductions of lentiviral particles containing *Nr1d1*-shRNA or *Nr1d1*-cDNA, respectively. The expression levels of *Nr1d1* in heart-derived Sca-1^+^CD31^−^ cells and MCM (Mouse Cardiac Myocytes) samples were detected using qRT-PCR and Western blotting. The results showed that the mRNA levels of *Nr1d1* were reduced by 90.9% (O-Sca-1^+^CD31^−^ cells, *p* < 0.0001) and 82.1% (MCM cells, *p* = 0.0007) in the *Nr1d1* knockdown cells (Figure 1B,C). The protein level of NR1D1 in MCM cells was reduced by 35.0% when compared to the control cells (Figure 1D, *p* = 0.0011). Overexpression of *Nr1d1* in Y-Sca-1^+^CD31^−^ and MCM cells increased the mRNA abundance by 5.133-fold (*p* = 0.0003) and 2.45-fold (*p* < 0.0001), respectively (Figure 1B,C). The protein level of NR1D1 rose by 1.25-fold in MCM cells compared to the control (Figure 1D, *p* = 0.0042), while the protein level of NR1D1 in Sca-1^+^CD31^−^ cells could not be detected because the cell number was not sufficient for Western blotting analysis.

### 2.3. Impact of Nr1d1 Knockdown on Cell Plasticity

In order to investigate the potential involvement of *Nr1d1* in the plasticity and senescence of Sca-1^+^CD31^−^ cells, we examined their abilities to differentiate into cardiomyocyte, smooth muscle, and endothelial lineages when *Nr1d1* is depleted in O-Sca-1^+^CD31^−^ cells transduced with lentivirus particles containing *Nr1d1*-shRNA. The results revealed that there were no significant differences in the numbers of differentiated cells expressing cTn (Figure 2A), α-SMA (Figure 2B), or VWF (Figure 2C) between the control and *Nr1d1*-shRNA-transfected cells, as evaluated by average fluorescence intensity on a single cell basis. These data indicate that downregulation of *Nr1d1* does not affect the differentiation of O-Sca-1^+^CD31^−^ cells.

### 2.4. The Effect of Nr1d1 Knockdown on Cell Senescence

Besides the *Nr1d1* irrelevant role in the differentiation potential, we further assessed whether *Nr1d1* participates in the proliferation potential of O- Sca-1^+^CD31^−^ cells. The results showed that the cell proliferation rate was 26.4 ± 0.91% for the *Nr1d1* knockdown O-Sca-1^+^CD31^−^ cells and 21.93 ± 1.19% for the control samples (*p* = 0.0067, Figure 3A) suggesting that depletion of *Nr1d1* increased cell proliferation. Moreover, *Nr1d1* knockdown promoted cell proliferation of the MCM cells (*p* = 0.0001, Figure 3B), indicating that the function of *Nr1d1* is related to cell proliferation. Meanwhile, the apoptosis rate was decreased to 25.27 ± 2.92% in *Nr1d1* knockdown O-Sca-1^+^CD31^−^ cells, compared to 31.38 ± 1.61% in the control cells (*p* = 0.0377, Figure 3C) suggesting that *Nr1d1* knockdown also inhibited apoptosis (*p* = 0.0025, Figure 3D). These results indicate that downregulation of *Nr1d1* promotes cell proliferation and inhibits apoptosis, which may be related to cellular aging.

Next, we assessed cell senescence after *Nr1d1* knockdown. Senescence-associated beta-galactosidase (SA-b-gal) activity displayed significantly lower counts in the *Nr1d1* knockdown group (*p* = 0.0382, Figure 3E). Furthermore, cell cycle inhibition seems to be associated with cellular senescence because the proportion of cells in the G0/G1 phase was 42.00 ± 1.16% in *Nr1d1*-knockdown O-Sca-1^+^CD31^−^ cells and 62.44 ± 2.31% in the control samples (*p* < 0.0001). In *Nr1d1*-knockdown O-Sca-1+CD31 and control cells, cell proportions in the S phase were 47.103.07% and 32.611.40% (*p* 0.0001), and those in the G2/M phase were 10.891.91% and 4.942.46% (*p* = 0.0161, Figure 3F). These findings reveal that downregulation of *Nr1d1* reverses aging in O-Sca-1^+^CD31^-^ cells and promotes more cells escaping the G0/G1 phase, thereby increasing the S- and G2/M-phase fractions.

Interestingly, *Nr1d1* knockdown in MCM cells promotes cell escape from the G0/G1 phase (Figure 3G). SR8278 is an antagonist of *Nr1d1* and inhibits *Nr1d1* expression (Appendix A, *p* = 0.0431). MCM cells treated with SR8278 exhibit similar effects, including increased cell proliferation (*p* < 0.0001), inhibition of apoptosis (*p* = 0.0012), more cells escaping the G0/G1 phase (*p* = 0.0006), and reversed cell aging (*p* = 0.0003) (Appendix A).

### 2.5. The Effect of Nr1d1 Overexpression on Cell Plasticity and Senescence

The potential of Y- Sca-1^+^CD31^−^ cells to differentiate into cardiomyocyte, smooth muscle, and endothelial lineages was assessed upon transduction with *Nr1d1*-cDNA or a control vector. There were no significant differences in the numbers of differentiated cells expressing cTn (Figure 4A), α-SMA (Figure 4B), and VWF (Figure 4C) between the control and *Nr1d1*-cDNA transfected cells, as evaluated by the average fluorescence intensity on a single cell basis.

We further examined the effect of *Nr1d1* overexpression on senescence-related factors. Cell proliferation rates were 16.03 ± 1.33% with *Nr1d1* overexpression in Y-Sca-1^+^CD31^−^ cells and 30.2 ± 2.16% in the control cells (*p* = 0.0006, Figure 5A), suggesting that upregulation of *Nr1d1* inhibited cell proliferation. *Nr1d1* overexpression in MCM cells had similar effects with inhibited cell proliferation (*p* < 0.0001, Figure 5B). Moreover, *Nr1d1* overexpression promoted apoptosis with 13.17 ± 2.73% apoptotic cells in *Nr1d1*-overexpressed Y-Sca-1^+^CD31^−^ cells and 6.07 ± 2.01% in the control cells (*p* = 0.0222, Figure 5C). *Nr1d1* overexpression in MCM also promoted apoptosis (*p* = 0.0250, Figure 5D). Furthermore, senescence-associated beta-galactosidase (SA-b-gal) activity displayed significantly higher counts after *Nr1d1* overexpression (*p* = 0.0010, Figure 5E). The percentage of cells in the G0/G1 phase was 75.38 ± 12.08% in the *Nr1d1*-overexpressed Y-Sca-1^+^CD31^−^ cells and 35.43 ± 5.43% in the control cells (*p* < 0.0001). The percentages in the S phase were 21.51 ± 5.17% and 44.57 ± 1.58% (*p* = 0.0021) and in the G2/M phase were 8.78 ± 4.03% and 19.82 ± 3.71%, in the *Nr1d1*-overexpressed Y-Sca-1^+^CD31^−^ and the control cells, respectively (Figure 5F). These results indicate that *Nr1d1* overexpression arrests cells in the G0/G1 phase, consequently inhibiting cells in the S and G2/M phases. Similarly, the percentage of cells arrested in the G0/G1 phase increased in the MCM cells after *Nr1d1* overexpression (Figure 5G).

In addition, we also treated MCM cells with GSK4112, an agonist of NR1D1. The results revealed that the cell proliferation of MCM cells was inhibited. At the same time, the proportion of apoptotic cells increased, cell cycle entry was slowed, and the proportion of senescent cells increased (Appendix A).

Overall, the above results indicate that *Nr1d1* is involved in the regulation of cell senescence.

### 2.6. The Molecular Mechanism Underlying the Regulation of Sca-1^+^CD31^−^ Cells Aging by Nr1d1

To investigate the mechanism by which *Nr1d1* regulates the Sca-1^+^CD31^−^ cell plasticity, gene expression data were obtained using a microarray comparing the *Nr1d1*-cDNA-transduced Sca-1^+^CD31^−^ cells and negative control. A total of 22 differentially expressed genes were identified, with a false discovery rate < 0.05 and fold change ≥ 2. Hierarchical clustering of differential genes was performed (Figure 6A). Microarray and qPCR showed similar *Nr1d1* expression levels. The *Nr1d1* levels were significantly upregulated in the *Nr1d1*-cDNA-infected Sca-1^+^CD31^−^ cells. The interaction relationship among differentially expressed genes reveals that *Nr1d1* directly interacts with *Nr4a3* and indirectly regulates the gene expression patterns in the Sca-1^+^CD31^−^ cells (Figure 6B). The expression of differentially expressed genes after *Nr1d1* knockdown and overexpression in Sca-1^+^CD31^−^ and MCM cells were further assessed by real-time PCR. The results showed that *Nr1d1* knockdown in O-Sca-1^+^CD31^−^ cells decreased the expression of *Dpt* (−1.926 ± 0.21%, *p* < 0.0001) and *Serpina3* (−1.951 ± 0.09%, *p* < 0.0001) and increased the expression of *Nr4a3* (1.363 ± 0.46%, *p* = 0.0079), *Il11* (1.219 ± 0.09%, *p* = 0.0001), and *Polr2k* (1.344 ± 0.19%, *p* = 0.0013), but did not affect the expression levels of other differentially expressed components (Figure 6C). On the other hand, overexpression of *Nr1d1* in Y- Sca-1^+^CD31^−^ cells increased the expression of *Dpt* (3.931 ± 2.46%, *p* = 0.0153) and *Serpina3* (1.897 ± 0.77%, *p* = 0.0171) and decreased the expression of *Nr4a3* (−3.028 ± 1.57%, *p* = 0.0070), *Il11* (−5.856 ± 1.32%, *p* < 0.0001), and *Polr2k* (−2.173 ± 0.27%, *p* < 0.0001), and did not affect the expression levels of other differentially expressed components (Figure 6D). The gene expression related to *Nr1d1* knockdown or overexpression in MCM cells was also examined. Among the genes we investigated, only *Nr4a3* was 1.84 ± 0.47 fold significantly upregulated with *Nr1d1* knockdown (*p* = 0.0091, Figure 6E), and 52.15% downregulated with *Nr1d1* overexpression when compared to the control (*p* = 0.0015, Figure 6F). Meanwhile, *Serpina3* was 50.05% downregulated in MCM cells transfected with *Nr1d1* shRNA (*p* = 0.0010, Figure 6E) and 1.94 ± 0.56 fold upregulated in MCM cells transfected with *Nr1d1*-cDNA when compared to the control (*p* = 0.0143, Figure 6F). Since the knockdown of *Nr1d1* caused significant upregulation of *Nr4a3* expression, we used the GCBI cloud platform to predict the interaction proteins of NR1D1. The results revealed that *Nr4a3* was closely related to *N**r1**d1* (Figure 6B).

To further elucidate whether *Nr1d1* directly regulates the expression of *Nr4a3*, we examined the binding of *Nr1d1* to the promoter region of *Nr4a3* using a luciferase reporter assay. The results showed that *Nr4a3* promoter activity was enhanced by 2.24 ± 0.46-fold (*p* < 0.0001) (Figure 7A). On the other hand, expression of a *Serpina3* promoter was increased by *Nr4a3* (*p* < 0.0001, Figure 7B). The data suggest that *Nr1d1* binds to the promoter of *Nr4a3* and represses the expression of *Nr4a3*, and subsequently *Nr4a3* binds to the promoter of *Serpina3* and represses the expression of *Serpina3*. 

To test this hypothesis, we knocked out *Nr4a3* using CRISPR/Cas9 and knocked down *Nr1d1* by *Nr1d1*-shRNA lentivirus in MCM cells. The results showed that *Nr4a3* knockout counteracted the significant proliferation that occurred due to downregulated *Nr1d1* in the MCM cells (*p* < 0.0001) (Figure 7C). Similarly, when *Nr4a3* was depleted, the number of senescent cells in *Nr1d1*-knockdown cells increased significantly (*p* = 0.0012), as measured by the senescence-associated -galactosidase assay (Figure 7D). To further verify the relationship between *Nr1d1*, *Nr4a3*, and *Serpina3*, we examined the expression of *Serpina3* in MCM cells with *Nr1d1*-knockdown or with *Nr1d1*-knockdown and *Nr4a3*-knockout. *Serpina3* expression levels were significantly increased in cells with both *Nr1d1*-knockdown and *Nr4a3* knockout (*p* = 0.0037, Figure 7E). These results confirm that *Nr1d1* controls the expression of *Nr4a3*, which has the ability to regulate cellular senescence through *Serpina3*.

## 3. Discussion

Aging affects the fundamental properties of O-Sca-1^+^CD31^−^ cells causing drastic changes of Sca-1^+^CD31^−^ cells including decreased differentiation ability, proliferation capacity, and increased apoptosis [14]. NR1D1 is a ligand-sensitive transcription factor that can promote BMSC (bone marrow mesenchymal stem cell) aging with inhibited cell proliferation and osteogenesis [17]. In line with this, our results revealed that the abundance of *Nr1d1* affects O- Sca-1^+^CD31^−^ cell proliferation and apoptosis. Elevated levels of *Nr1d1* in Y-Sca-1^+^CD31^−^ cells triggered apoptosis and regulated the cell cycle to arrest cells in the G0/G1 phase, which then inhibited cell proliferation. This indicates that *Nr1d1* promotes heart-derived Sca-1^+^CD31^−^ aging by hindering cells from entering the cell cycle and thus cell proliferation. Reduced *Nr1d1* levels in O-Sca-1^+^CD31 cells, on the other hand, inhibited cell apoptosis, regulated cell cycle stages to escape the G0/G1 phase, and increased the volumes of the S- and G2/M-phases, promoting cell proliferation. However, changes in *Nr1d1* abundance did not affect the cell differentiation potential. *Nr1d1* may indirectly affect the function and aging of the mouse heart by regulating the proliferation and senescence of Sca-1^+^CD31^−^ cardiac cells. 

Microarray analysis identified 22 differentially expressed genes when *Nr1d1* is upregulated. Among them, many genes have not been previously identified as *Nr1d1*-responsive. Only *Nr4a3* and *Serpina3* were reported to be directly or indirectly related to *Nr1d1*.

NR4A3 (also known as NOR1) belongs to the NR4A subfamily of nuclear receptors and is a constitutively active transcription factor regulating cellular proliferation, differentiation, inflammation, and apoptosis [18,19]. Overexpression of *Nr4a3* reduced VSMC apoptosis [20] and increased cell proliferation, with *cyclin D1* and *D2* as *Nr4a3*-target genes [19,21]. *Nr4a3* deletion in hematopoietic stem cells (HSCs) accelerated atherosclerosis formation [18]. *Serpina3* is a member of the serine proteinase inhibitor gene family and is highly expressed in HSCs [22]. *Serpina3* has been shown to be involved in anti-adipogenesis [23] and anti-inflammation, and is associated with a number of human diseases [24]. *Serpina3* was identified as an *ApoA4*-regulated gene transcriptionally via nuclear receptors *Nr4a3* and *Nr1d1* in hepatocytes [24]. A recent study confirmed that the downstream apoptosis trigger was independent of *p53* by REV-ERB agonists [13]. The *Nr1d1* overexpression in Young Sca-1^+^CD31^−^ cells displayed the characteristics of aged cells. Expression analysis indicated that *Nr1d1* increased the expression of *Serpina3* but decreased the expression of *Nr4a3*. The luciferase reporter assay demonstrated that the expression of the *Nr4a3* promoter was enhanced by *Nr1d1*, and the expression of the *Serpina3* promoter was increased by *Nr4a3*. In addition, knockout of *Nr4a3* reversed the proliferation and senescence of MCM cells due to knockdown of *Nr1d1* and upregulated the expression of *Serpina3*. Therefore, we infer that *Nr1d1* binds to the promoter of *Nr4a3* and represses the expression of *Nr4a3*, and subsequently, the downregulated expression of *Nr4a3* promotes the expression of *Serpina3* due to the decreased binding of *Nr4a3* to the promoter of *Serpina3*.

Our findings suggest that *Nr1d1* can directly regulate gene expression in various types of cells. However, additional studies on the molecular mechanism by which *Nr1d1* regulates Sca-1^+^CD31^−^ cell aging are required. It was reported that *Nr1d1* plays a key role in circadian rhythm [2]. Our microarray data, on the other hand, revealed that *Nr1d1* did not regulate the circadian components in the Sca-1^+^CD31^−^ cells. This indicates that the role of *Nr1d1* in circadian rhythm may be cell-specific. Although some studies suggest that *Rev-erb* agonists are specifically lethal to cancer cells and have no effect on the viability of normal cells or tissues [15], *Rev-erb* agonist SR9009 treatment inhibits post-myocardial infarction mortality and improves cardiac function by modulating inflammation and remodeling processes [25]. In contrast, our data showed that increased *Nr1d1* expression in heart Sca-1^+^CD31^−^ cells may contribute to the reduced cell repair capacity. The intrinsic determinants of cellular senescence with higher *Nr1d1* expression promote cell death and impair cell growth, compromising myocyte loss and decreasing cardiac function with age. These results strongly indicate that the effect on the heart should be taken into account when the pharmacological modulation of circadian machinery is used as a cancer treatment.

## 4. Materials and Methods

### 4.1. Experimental Animals and Cell Culture

Mice were purchased from SLAC Ltd. (Changsha, China). (SCXK(Xiang)2011-0003). Animals were maintained in the Guangxi Normal University Laboratory Animal Center and handled in accordance with the institution’s guidelines. All experimental protocols were approved by the Guangxi Normal University Animal Management Committee of Guangxi S&T Department (Approval Number: 20190313-003). FACS-sorted (Becton, Dickinson and Company, Franklin Lakes, NJ, USA) Lin^−^CD45^−^Sca-1^+^CD31^−^ were used for experiments. Young (Y, 2 months old) and old (O, 22 months old) mice Sca-1^+^CD31^−^ cells, respectively, were prepared and cultured as previously described [11,14]. 

The Mouse Crdiac Myocytes (MCM) cell line was isolated from postnatal day 2 mouse heart, and can be secondarily cultivated [26]. It was purchased from BeNa Culture Collection (Beijing, China) and cultured in DMEM containing 10% FBS, 100 μg/mL streptomycin, and 100 U/mL penicillin at 37 °C under 5% CO_2_. The MCM cells were passaged after 3 days. To study the effects of Nr1d1 agonists, MCM cells were incubated in DMEM without FBS for 16 h to deplete intracellular heme concentration and then switched to DMEM supplemented with either DMSO or Nr1d1 agonist GSK4112 (10 μM) or antagonist SR8278 (10 μM) for 6 h.

### 4.2. Nr1d1lentivirus Vectors Constructs and Stable Transfection

For *Nr1d1* overexpression, full-length murine *Nr1d1* cDNAs were obtained by PCR using the primers 5′-tatcgaattc (EcoRI) ATGACGACCCTGGACTCCA-3′ (forward) and 5′-GGATCCGCGG CCGCTTCTAGAtatcggatcc(BamHI)-3′(reverse). The amplified *Nr1d1* cDNAs fragments were digested at EcoRI/BamHI and then cloned into CMV-MCS-IRES-EGFP lentiviral cDNA vector (Hanbio, Shanghai, China).

For *Nr1d1* knockdown, oligonucleotides (GCAAGGCAACACCAAGAA TGT) targeting *Nr1d1* were used to clone short hairpin (sh)RNA into the hU6-MCS-PGK-EGFP lentiviral RNAi vector (Hanbio). The recombinant overexpression and knockdown lentiviral vectors were produced by co-transfection of 293T cells with the pSPAX2, pMD2G, and pHBLV plasmids using LipoFiter reagent (Hanbio). Lentivirus-containing supernatant was collected 48 h after transfection and passed through a 0.45-μm cellulose acetate filter (Millipore, Billerica, MA, USA). Recombinant lentiviruses were concentrated by ultracentrifugation for 2 h at 72,000× *g*.

MCM cells were transfected with the lentiviral *Nr1d1*cDNA vector, lentiviral *Nr1d1* RNAi vector, or empty vector encoding EGFP, respectively, using polybrene (5 μg/mL, Hanbio). Selection was initiated in medium containing 2 μg/mL puromycin (Invitrogen, Waltham, MA, USA). After selection, a stable *Nr1d1* transfectant overexpression, knockdown, and the control cell line were established.

### 4.3. Nr4a3 Knockout and Nr1d1 Knockdown

The sgRNA sequences of mouse *Nr4a3* (mm10/GRCm38) exons were retrieved using the CHOPCHOP website (http://chopchop.cbu.uib.no/, accessed on 13 October 2022). sgRNA without off-target and upstream of the exon was selected as the CRISPR/Cas9 target sequence (Appendix A). The sgRNA plasmid pTianJ2-*Nr4a3* was constructed using the pTianJ2 plasmid. The constructed plasmid was cotransferred into MCM cells with the pST1374-Cas9-ZFNLS WT plasmid. Monoclonal cells were obtained by an infinite dilution method after puromycin screening. DNA sequencing of the target gene was performed to select the *Nr4a3*-knockout cell line (Appendix A).

MCM-*Nr4a3*-knockout cells were transfected with the lentiviral *Nr1d1* RNAi vector to obtain cells with both *Nr1d1* knockdown and *Nr4a3* knockout. After two generations of cell passages, GFP positive cells were sorted by flow cytometry (Becton, Dickinson and Company, Franklin Lakes, NJ, USA).

### 4.4. Cell Differentiation and Immunofluorescence Analysis

Two days after the viral infection, lentivirus-infected and negative control Sca-1^+^CD31^−^ cells were cultured in cardiomyocytes, smooth muscle cells, and endothelial differentiation induction medium for 14 days. Then, they were examined for the expression of cTnI, α-SMA, and VWF by immunocytochemistry. The experiment details as previously described [11,14].

### 4.5. Analysis of Cell Cycle, Proliferation, and Apoptosis

Y-/O-Sca-1^+^CD31^−^ cells were transduced with *Nr1d1*-shRNA, *Nr1d1* cDNA, and negative control lentiviral particles. After 48 h, cell apoptosis was detected by flow cytometry (Becton, Dickinson and Company, Franklin Lakes, NJ, USA) using an Annexin V-phycoerythrin (PE) Apoptosis Detection kit (Invitrogen, Carlsbad, CA, USA). The percentage of apoptotic cells was defined as the sum of annexin V-PE single-positive and annexin V-PE/7-aminoactinomycin D (7-AAD) double-positive cells. The 5-ethynyl-2′-deoxyuridine (EdU) assay was performed to assess cell proliferation 48 h after lentiviral infection by incubating the cells in a 10 μmol/L EdU solution (RiboBio, Guangzhou, China) for 2 h followed by flow cytometry analysis. Cell cycle analysis was performed using the 7-AAD Flow Cytometry Assay kit (Ebioscience, San Diego, CA, USA) according to the manufacturer’s instructions. These methods have been described in detail in previous work [11,14]. Cell apoptosis and cell cycle are performed on MCM cells as in Sca-1^+^CD31^−^ cells. Proliferation of MCM cells was detected by MTT assay (Solarbio, Beijing, China).

### 4.6. Microarray Gene Expression Data

The Affymetrix Mouse Genome 2.0 Microarray (Santa Clara, CA, USA) was utilized to analyze gene expression of samples by high-throughput technologies. The experiments detail and data access were described in previously published studies [27,28]. Corresponding CEL files are publicly available on the Gene Expression Omnibus database (accession numbers GSE43556, GSM1024592-94, and GSE7196).

### 4.7. Quantitative Real-Time RT-PCR and Droplet Digital PCR

Total RNA and cDNA were prepared using a kit (Promega, Madison, WI, USA) according to manufacturer’s instructions. The details of real-time RT-PCR (Roche, Basel, Switzerland) and Droplet digital PCR(ddPCR) (Bio-Rad, Hercules, CA, USA) were described previously [29]. The primers were listed in Appendix A.

### 4.8. Luciferase Assay

The promoter sequence (2000 bp) upstream of the transcriptional start site of mouse *Nr4a3* and *Serpina3* was cloned into luciferase reporter plasmid and verified with direct sequence, respectively, as m-*Nr4a3*-pro+pRL-TK and m-*Serpina3*-pro+pRL-TK. The MCM cells (2 × 10^5^cells/well in 24-well plates) were transiently transfected with pGL3 or *Nr1d1* plasmids (m-*Nr1d1*) and pRL-TK plasmid (Promega, Madison, WI, USA) using lipofectamine 3000. Cells were lysed 24 h after transfection and assayed for firefly and Renilla luciferase activity using the Dual-Luciferase reporter system (Promega, Madison, WI, USA). The data are expressed as the ratios of firefly to Renilla activity.

### 4.9. Statistical Analysis

Data analysis was performed using GraphPad Prism 6 software (GraphPad software, San Diego, CA, USA). Two-sided *p*-values were calculated, and *p* < 0.05 was considered statistically significant.

## 5. Conclusions

This study revealed an important role of *Nr1d1* and the underlying mechanism in cardiac aging. High expression of *Nr1d1* in cardiac-derived Sca-1^+^CD31^-^ cells and MCM causes cell cycle arrest in the G0/G1 phase, inhibits proliferation, promotes apoptosis and senescence, and thus promotes cardiac senescence. Knockdown of *Nr1d1* expression in Sca-1^+^CD31^−^ cells and MCM promoted cell proliferation and reduced cellular senescence. *Nr1d1* induces *Serpina3* expression via *Nr1d1* interaction with *Nr4a3*, thus promoting cell senescence. The finding will establish the functional relationship between cardiac senescence and *Nr1d1* abnormal expression, reveal the molecular pathway of *Nr1d1* in the regulation of cardiac senescence and its functional degeneration, and provide new targets and strategies for the diagnosis and treatment of age-related heart diseases.

## Figures and Tables

**Figure 1 ijms-23-12455-f001:**
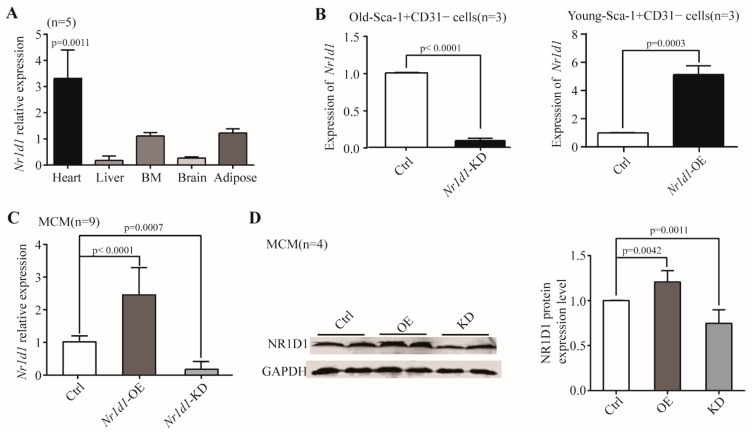
The expression of *Nr1d1*. (**A**) Real-time RT-PCR was used to identify the mRNA expression level of *Nr1d1* in aged organs. Each bar represents the fold change of gene expression in aged vs. young mice (*n* = 5). The expression levels were normalized by *G**apdh* and the expression level in young mice was used as a calibrator to calculate the fold change. A two-way ANOVA was applied for the analysis. *p* < 0.05 was considered statistically significant; (**B**) DD-PCR analysis of *Nr1d1* mRNA expression in Sca-1^+^CD31^−^ cells (*n* = 3). The left bar graph shows a knockdown of *Nr1d1* in the old Sca-1^+^CD31^−^ cells transduced with *Nr1d1*-shRNA lentiviral particles (*Nr1d1*-knockdown), while the right bar graph shows overexpression of *Nr1d1* in the young Sca-1^+^CD31^−^ cells transduced with *Nr1d1*-cDNA lentiviral particles (*Nr1d1*-overexpression). The control cell was transduced with an empty lentivector. A two-tailed unpaired *t*-test was applied for analysis. *p* < 0.05 was considered statistically significant; (**C**) real-time RT-PCR analysis of *Nr1d1* overexpression in MCM cells transduced with *Nr1d1*-cDNA lentiviral particles (*n* = 9). One-way ANOVA was used for analysis. *p* < 0.05 was considered statistically significant; (**D**) immunoblot analysis of NR1D1 protein levels in MCM cells transduced with *Nr1d1*-cDNA lentiviral particles (*n* = 4). Overexpression and knockdown of *Nr1d1* were performed using lentiviral transduction of *Nr1d1*-cDNA and *Nr1d1*-shRNA, respectively. The control was transduced with an empty lentivector in MCM cells. The data are from three independent experiments and are presented as mean ± SD. Two-way ANOVA was used to analyze the data. *p* < 0.05 was considered statistically significant. Statistical analysis was implemented by GraphPad Prism 5. Ctrl: control, KD: knockdown, OE: overexpression.

**Figure 2 ijms-23-12455-f002:**
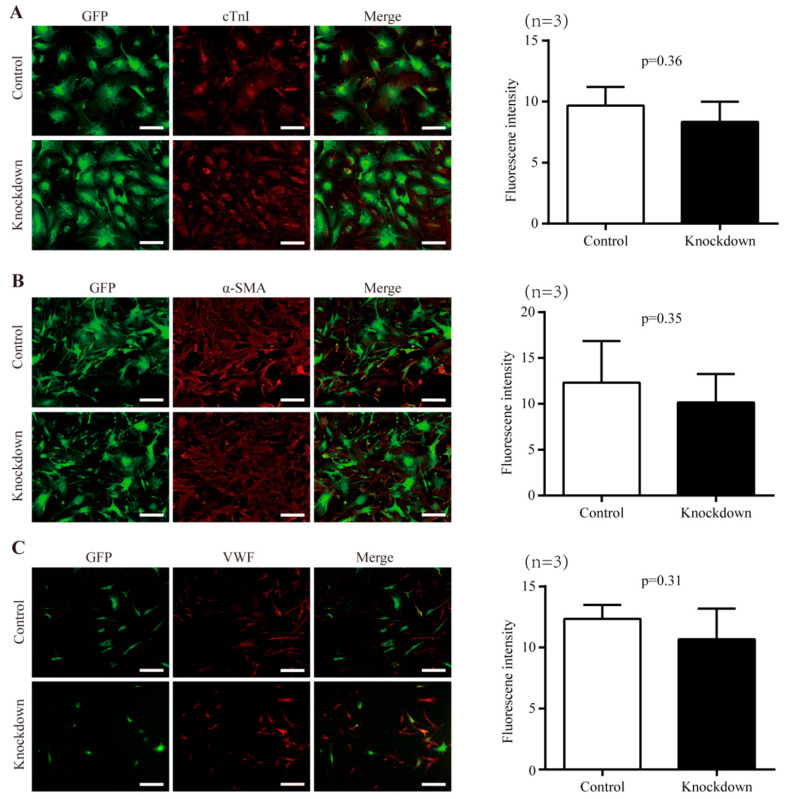
The effect of *Nr1d1* knockdown on cell plasticity. (**A**–**C**) cell differentiation of the old Sca-1^+^CD31^−^ cell after the *Nr1d1* knockdown compared to the negative control group. The left panel shows representative immunofluorescence images stained with cardiomyocyte-specific marker cTnI (red) (**A**), smooth muscle marker α-SMA (**B**), and endothelial cell marker VWF (**C**). Sca-1^+^CD31^−^ cells were marked with GFP merged with DyLight^®^ 550 (red). Scale bars: 50 μm. The bar graph on the right panel depicts the grayscale values of DyLight^®^ 550 within the GFP region as determined by ImageJ and Photoshop, with data from three independent experiments (*n* = 3) presented as mean ± SD. Two-tail unpaired *t*-test was used for the analysis method, and *p* < 0.05 was considered statistically significant.

**Figure 3 ijms-23-12455-f003:**
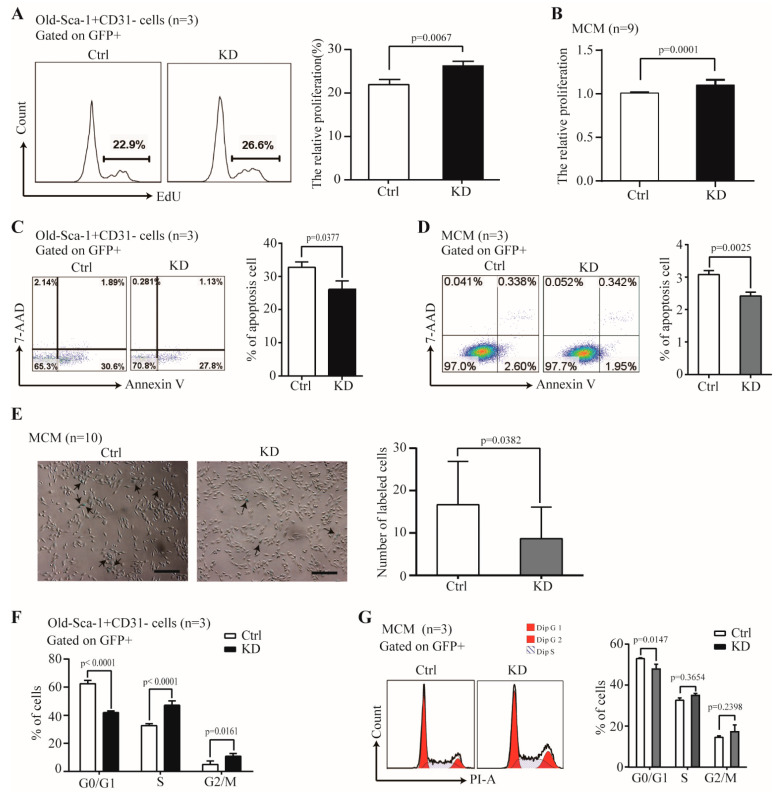
The effect of *Nr1d1* knockdown on cell senescence. (**A**) cell proliferation of old Sca-1^+^CD31^−^ cells transduced with a *Nr1d1*-shRNA lentivirus as indicated by the embedding of EdU. The flow cytometric analysis of representative Edu embedding is shown in the left panel, and the bar graph on the right panel is a statistical plot of EdU embedding. The data were from three independent experiments (*n* = 3). Two-tail unpaired *t*-test was applied for statistical analysis, *p* < 0.05 was considered significant; (**B**) MTT analysis of cell proliferation of MCM transduced with a lentiviral *Nr1d1*-shRNA. The data were from three independent experiments, and each experiment had three replicates (*n* = 9). A two-tail unpaired *t*-test was used for statistical analysis, and *p* < 0.05 was considered significant. (**C**,**D**) Apoptosis rate of the old Sca-1^+^CD31^−^ cells (**C**) and MCM (**D**) transduced with lentiviral *Nr1d1*-shRNA analyzed by FCM with annexin V-PE/7-AAD staining. The left panel is a representative flow cytometric scatter plot, and the right panel is a bar graph of apoptosis statistics. The data were from three independent experiments (*n* = 3). A two-tail unpaired *t*-test was applied for statistical analysis, and *p* < 0.05 was considered significant; (**E**) senescence-associated beta-galactosidase staining of MCM transduced with a lentiviral *Nr1d1*-shRNA. Control cells were transduced with an empty lentivector. The image on the left panel is a representative field of view, scale bars: 50 μm. The bar graph on the right panel shows the number of cells that were stained blue in each random field of view. Each experiment was repeated three times, and 10 random fields of view were counted (*n* = 10). A two-tail unpaired *t*-test was used for statistical analysis, and *p* < 0.05 was considered significant. (**F**,**G**) Cell cycle analysis by FCM for the old Sca-1^+^CD31^−^ (**F**) and MCM (**G**) cells transduced with lentiviral Nr1d1-shRNA. The data were collected from three independent experiments (*n* = 3), two-way ANOVA was applied for statistical analysis, and *p* < 0.05 was considered significant. All data in the bar graph were presented as mean ± SD. Ctrl: control, KD: knockdown.

**Figure 4 ijms-23-12455-f004:**
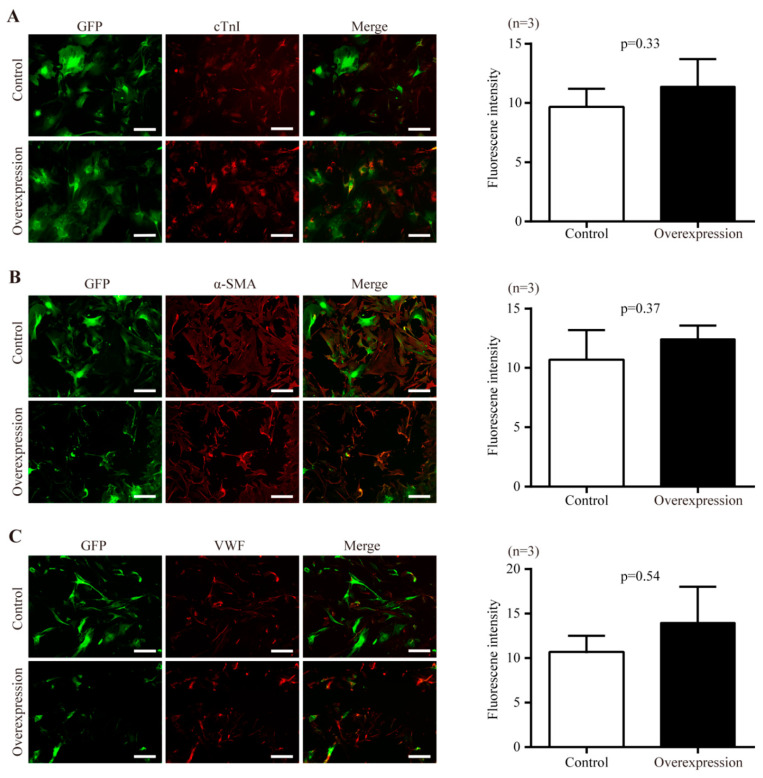
The effect of *Nr1d1* overexpression on cell plasticity. (**A**–**C**) differentiation of young Sca-1^+^CD31^−^ cells after *Nr1d1* overexpression compared to the negative control group. The left panel shows representative immunofluorescence images stained with cardiomyocyte-specific marker cTnI (**A**) (red), smooth muscle marker α-SMA (**B**), and endothelial cell marker VWF (**C**). Sca-1^+^CD31^−^ cells were marked with GFP merged with DyLight^®^ 550 (red). Each experiment was repeated three times, and each image represents a single result. Scale bars, 50 μm. The bar graph on the right panel depicts the grayscale values of DyLight^®^ 550 within the GFP region as determined by ImageJ and Photoshop, with data from three independent experiments (*n* = 3) presented as mean SD. A two-tail unpaired *t*-test was used for statistical analysis, and *p* < 0.05 was considered significant.

**Figure 5 ijms-23-12455-f005:**
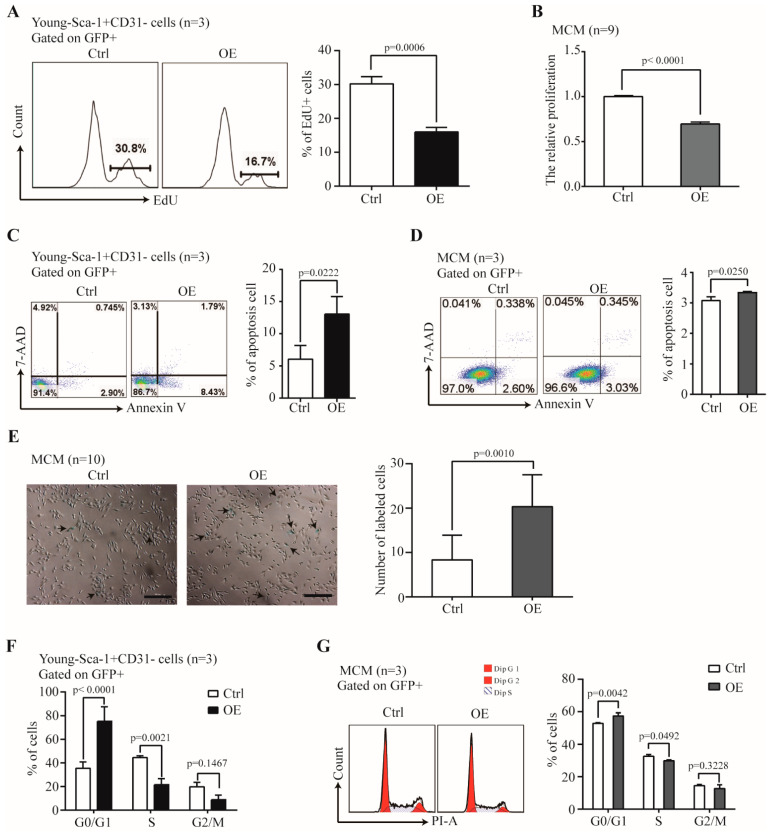
The effect of *Nr1d1* overexpression on cell senescence. (**A**) Cell proliferation of young Sca-1^+^CD31^−^ cells transduced with *Nr1d1*-cDNA lentiviral as indicated by the embedding of EdU. The left panel shows a flow cytometric analysis of a representative Edu embedding, and the right panel shows a statistical plot of Edu embedding. The data were from three independent experiments (*n* = 3). A two-tail unpaired *t*-test was applied for statistical analysis. *p* < 0.05 was considered significant. (**B**) MTT analysis of cell proliferation of MCM transduced with lentiviral *Nr1d1*-cDNA was detected. The data were from three independent experiments, and each experiment had three replicates (*n* = 9). A two-tail unpaired *t*-test was used for statistical analysis, and *p* < 0.05 was considered significant. (**C**,**D**) Apoptosis rate of young Sca-1^+^CD31^−^ cells (**C**) and MCM (**D**) transduced with lentiviral *Nr1d1*-cDNA analyzed by FCM with annexin V-PE/7-AAD staining. The left panel is a representative flow cytometric scatter plot, and the right panel is quantification of apoptosis. The data were from three independent experiments (*n* = 3). A two-tail unpaired *t*-test was applied for statistical analysis, and *p* < 0.05 was considered significant; (**E**) senescence-associated beta-galactosidase staining of MCM transduced with lentiviral *Nr1d1*-cDNA. Control was transduced with an empty lentivector. The image on the left panel is a representative field of view. The scale bars: 50 μm. The bar graph on the right panel shows the number of cells that were stained blue in each random field of view. Each experiment was repeated three times, and 10 random fields of view were counted (*n* = 10). A two-tail unpaired *t*-test was used for statistical analysis, and *p* < 0.05 was considered significant; (**F**,**G**) cell cycle analysis by FCM for young Sca-1^+^CD31^−^ cells (**F**) and MCM (**G**) cells transduced with lentiviral *Nr1d1*-cDNA. The data were collected from three independent experiments (*n* = 3), two-way ANOVA was applied for statistical analysis, and *p* < 0.05 was considered significant. All data in bar graph presented as mean ± SD. Ctrl: control, OE: overexpression.

**Figure 6 ijms-23-12455-f006:**
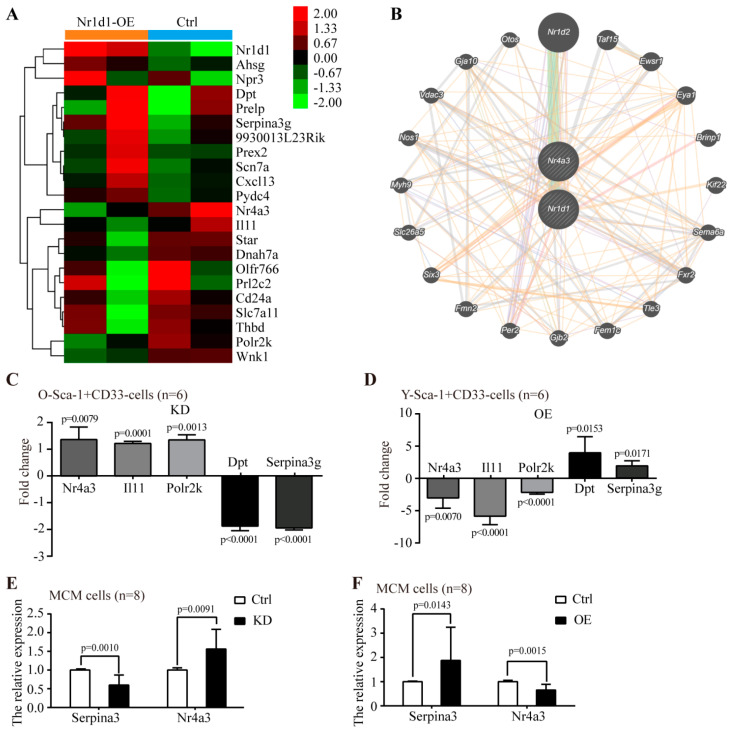
The effect of altered *Nr1d1* expression levels on gene expression profiles. (**A**) the differentially expressed genes identified by a microarray in Young Sca-1^+^CD31^−^ cells comparing the *Nr1d1*-cDNA-infected cells and negative control. The tree was based on log2 transformation of normalized probe signal intensity using hierarchical clustering. Red: upregulated gene expression; Green: downregulated gene expression. Every sample was tested twice. A total of 22 differentially expressed genes were identified via pairwise comparison. (**B**) Interaction relation of differentially expressed genes by generadar (https: //www.gcbi.com.cn/gcanalyze/html/generadar/index, accessed on 8 June 2020). (**C**,**D**) differentially expressed genes detected by real-time RT-PCR after *Nr1d1* knockdown (**C**) and overexpression (**D**) in Sca-1^+^CD31^−^ cells; data were from three independent experiments, and each experiment had two replicates (*n* = 6). (**E,****F**) Differentially expressed genes detected by real-time RT-PCR after *Nr1d1* knockdown (**E**) and overexpression (**F**) in MCM. Expression levels were normalized by *G**apdh*, and expression levels in the negative control were used as a calibrator to calculate fold changes. Calculated difference change based on a mean of three independent experiments, and each experiment had at least two replicates (*n* = 8). The ΔΔCT values were subjected to unpaired Student’s *t*-test implemented using Prism software. Bars above and below the *x*-axis show genes that are up- or downregulated, respectively. All data in bar graph presented as mean ± SD, and two-way ANOVA was used as statistical analysis, and *p* < 0.05 was considered significant. Ctrl: control, KD: knockdown, OE: overexpression.

**Figure 7 ijms-23-12455-f007:**
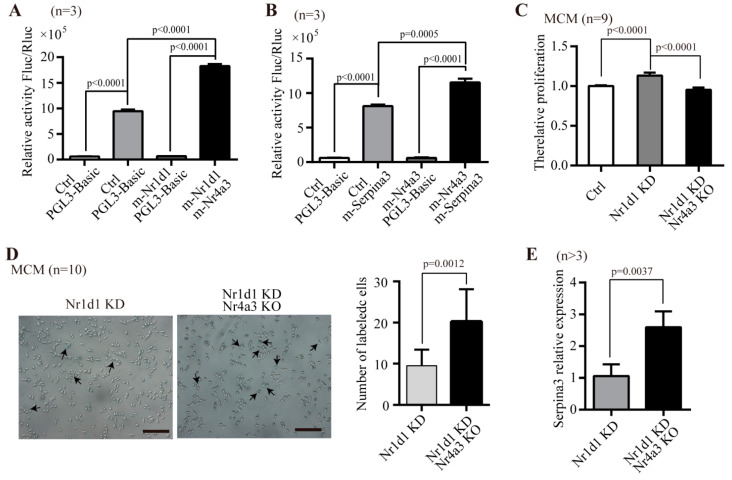
Regulatory mechanism of *Nr1d1*. (**A**,**B**) effect of *Nr4a3* and *Serpina3* on transcription of *Nr1d1* assessed by a Dual-Luciferase reporter system. MCM cells were transfected with *Nr4a3* luciferase reporter plasmid for 24 h, and incubated with or without the *Nr1d1* plasmid (**A**) or MCM cells were transfected with *Serpina3* luciferase reporter plasmid for 24 h, and incubated with or without the *Nr4a3* plasmid (**B**). Cells were lysed with lysis buffer, and the lysates were subjected to the luciferase activity assay, data were collected from three independent experiments (*n* = 3), two-way ANOVA was applied for statistical analysis, and *p* < 0.05 was considered significant. (**C**–**E**) knockdown of *Nr4a3* in MCM cells by CRISPR/Cas9 technique, followed by downregulation of *Nr1d1* using *Nr1d1*-shRNA lentivirus transduction; (**C**) the effect of *Nr4a3* knockout on cell proliferation in the MCM cells with *Nr1d1* knockdown. The data were from three independent experiments, and each experiment had three replicates (*n* = 9). One-way ANOVA was used for statistical analysis and *p* < 0.05 was considered significant; (**D**) the effect of *Nr4a3* knockout on cellular senescence (beta-galactosidase staining) in MCM cells with *Nr1d1* knockdown. The image on the left panel is a representative field of view, scale bars: 50 μm. The bar graph on the right panel shows the number of cells that were stained blue in each random field of view. Each experiment was repeated three times, and 10 random fields of view were counted (*n* = 10). Two-way ANOVA was used for statistical analysis, and *p* < 0.05 was considered significant. (**E**) The *Serpina3* mRNA expression levels were detected by real-time PCR on *Nr1d1* knockdown MCM cells versus on *Nr1d1* knockdown and *Nr4a3* knockdown MCM cells (*n* = 3 vs. 5); a two-tail unpaired *t*-test was used as an analysis method, *p* < 0.05 was considered statistically significant. All data in bar graph presented as mean ± SD. Ctrl: control, KD: knockdown, KO: knockout.

## Data Availability

Not applicable.

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
