# Peer review of "Nr1d1* Mediated Cell Senescence in Mouse Heart-Derived Sca-1^+^CD31^−^ Cells"

_ijms, 2022, doi:10.3390/ijms232012455_

Round 1
Reviewer 1 Report (New Reviewer)
I congratulate the authors for their work. The manuscript is well written. The abstract is pertinent and representative of the manuscript. The methodology is detailed. The results are well presented in blot diagrams and figures. The conclusions are valid and unbiased. The discussion is well written and covers the appropriate literature, and relevant citations have been provided. This manuscript will add new knowledge to the field.
Author Response
Thank you for your recognition of our manuscript and we will continue to work hard to do good research. We also wish you a great work performance. The manuscript has been invited to Professor Hongxia Zhao of the University of Helsinki(Finland) for English language revision.
Reviewer 2 Report (New Reviewer)
Pu et al. examined the functional role and the underlying molecular mechanism of Nr1d1 expression in cardiac senescence. Overexpression of Nr1D1 in cardiac-derived Sca-1+CD31- cells and mouse cardiomyocytes induced cell cycle arrest in the G0/G1 phase, inhibited proliferation, and promoted apoptosis and senescence. In agreement, knockdown of Nr1D1 expression in Sca-1+CD31- cells and mouse cardiomyocytes promoted cell proliferation and reduced cellular senescence. Moreover, Nr1d1 regulated Serpina3 expression via its interaction with Nr4a3 to promote cell senescence. The study is well designed, the results are highly interesting, a few issues should be clarified.
1. Detailed information about statistical tests should be provided. The results of the statistical tests should be reported in details. Bar graphs should be avoided. For guidelines see the following works: Weissgerber et al. eLife 2018;7:e36163; and Michel et al. J Pharmacol Exp Ther 2020;372:136-147.
2. Please describe the characteristics of the mouse cardiac myocyte cell line used in the present study.
3. The Authors suggested that, “The main effect of Nr1d1 may be related to the self-renewal ability of the heart…” (Discussion, lines 307-308). Which cell type may play a role in cardiac repair? Recent studies using novel mouse genetic engineering technology from independent labs provided evidence that Sca-1+ cells do not generate new cardiomyocytes during normal cardiac homeostasis or after injury (He et al. Circulation. 2020;142:275-291 and references within this review).
4. Sca-1+CD31+ cells are mentioned instead of Sca-1+CD31− cells in the 5th section of the manuscript (Conclusions, lines 451-454). Please clarify this issue.
Author Response
Thank you very much for reviewing our manuscript and giving great suggestions.
1) we have modified it according to the provided references.we have provided detailed information about statistical tests.
2)We have described he characteristics of the mouse cardiac myocyte cell line in the materials and methods section.
3)“The main effect of Nr1d1 may be related to the self-renewal ability of the heart. “This sentence does have some inappropriateness in the manuscript. Therefore, we have revised the sentence. “Nr1d1 may indirectly affect the function and aging of the mouse heart by regulating the proliferation and senescence of Sca-1+CD31− cardiac cells.”
Several studies have reported that Sca-1+ stem cells from adult heart can differentiate into cardiomyocytes(Song et al.J Cell Mol Med.2019; Meng et al. Genes (Basel).2020): smooth muscle like cells (Song et al.J Cell Mol Med.2019); endothelial(),adipocytes(Matsuura et al, J Biol Chem2004), etc. in vitro. Our previous study also found that Sca-1+CD31- cardiac-derived cells can differentiate into cardiac cell lineages (?-Smooth Muscle Actin, Von Willebrand Factor, Cardiac Troponin I) in vitro (Wu et al,Stem cell int. 2016), but due to limited experimental conditions, we were unable to perform in vivo experiments. Liang et al. found that the Sca-1+CD31- population increased after myocardial infarction, and transplantation of these cells significantly improved cardiac function by promoting angiogenesis (Wang et al. Stem Cells. 2006, Liang et al. Int J Cardiol. 2010). Stephanie et al.(Cardiovasc Res.2020) direct comparison of the paracrine effects of Sca-1+ cardiac progenitor cells and a heterogeneous population of mononuclear cardiac cells the supernatants of both cell types showed cardioprotective properties.In addition, genetic lineage tracing based on Sca-1 knock-in Cre lines showed that Sca-1+ cells mainly adopted an endothelial, but not cardiomyocyte, cell fate during homeostasis or after injury(Vagnozzi et al. Circulation. 2018; Zhang et al. Circulation. 2018; Neidig et al. Circulation. 2018; Tang et al. Circulation. 2018), It suggests that the beneficial effect from Sca-1+ cell transplantation is unlikely to be caused by direct cardiomyocyte differentiation, but more from angiogenesis and paracrine effects. Based on these studies and the ability of the Sca-1+CD31- subpopulation to differentiate vascular endothelial cells in vitro, we seem to infer that Sca-1+CD31- may be a progenitor or precursor cell of the vascular endothelium. Although many studies have demonstrated that these cardiac-derived stem cells do not manifest the ability to differentiate into cardiomyocytes in vivo, these studies do not seem to take into account the role of the cardiac microcirculation. Cardiac-derived stem cells are capable of differentiating into cardiomyocytes in vitro in isolation from the regulation of the cardiac microenvironment.
4)In the conclusion section, we incorrectly wrote Sca-1+CD31- as Sca-1+CD31+, which has now been corrected.
In addition, the full manuscript has been invited to Professor Hongxia Zhao of the University of Helsinki(Finland) for English language revision. The revised manuscript is attached.
This manuscript is a resubmission of an earlier submission. The following is a list of the peer review reports and author responses from that submission.
Round 1
Reviewer 1 Report
Pu et al functionally characterized the nuclear heme receptor, Nr1d1 in yound and aged endothelial precursor cells as well as mous cardiac myocytes. The data is of great interest since they reported a potential novel molecular mechanism. However some issues have to addressed:
- In section 2.6 additional experiments are required to further prove the implication of Nr4a3 in the regulation of the differentially expressed. Therefore, silencing or overexpression of Nr4a3 experiment are needed.
- Figure need to improved: p values should be added to the diagramms instead of astrixes. Information what the error bars indicates are required.
- there is an error in figure 1 b: I think that yuang stands for young?
Author Response
Thank you very much for reviewing our manuscript and giving great suggestions.
-In section 2.6, It is essential to add Nr4a3 knockdown or overexpression experiments to verify its role in regulation. This is consistent with the content of a Guangxi Natural Science Foundation that we applied for approval. We are going to test the effect on MCM after Nr4a3 knockdown by CRISPR/cas9, as well as validate the interaction with Nr1d1 and Serpina3. However, because of COVID-19 and the Chinese New Year, the implementation of the above work was slow. We will submit the results obtained for future publication of the manuscript. In addition, in the discussion of the manuscript, we cited "Effect of ApoA4 on SERPINA3 mediated by nuclear receptors NR4A1 and NR1D1 in hepatocytes" (Zhang et al. 2017). In this paper, they performed RNA interference of Nr1d1 and NR4A1 to verify the role with ApoA4 and SERPINA3 in hepatocytes.
-We have replaced the astrixes with p values in all figures and attached the p values to the text. All the error bars in Figure are presented as SD(mean ±SD), which we have added to the legend of figure.
-In Figure 1b, the spelling error was also corrected. In addition, we did another check of the manuscript to reduce spelling errors.
Thank you again for your review of our manuscript.
Best wish!
Reviewer 2 Report
In this manuscript Pu et al. analyze the involvement of Nr1d1 in cell proliferation, cell senescence cell apoptosis and cell plasticity. Further, in the absence of solid experimental data, they try to infer underlying mechanisms based on microarray data, leading to highly questionable conclusions.
Major points
- Abstract. According to the abstract the aim of the study was to deepen in the mechanism by which Nr1d1 modulates Sca-1+CD31- cells plasticity. However, in the present study Nr1d1 knockdown did not affect Sca-1+CD31- cells plasticity (Figure 2).
- Regardless of what the title of the manuscript states the authors neither show an effect of Serpina3 on Sca-1+CD31- cells senescence nor the involvement of Nr4a3 in SerpinA3-mediated cell senescence. Surprisingly, authors link Nr4a3 with an effect of serpinA3 in cell senescence, despite an inverse regulation of Nr4a3 and SerpinA3 expression is observed when Nr1d1 is silenced or overexpressed (Figure 6 and D), and Nr4a3 expression decreases when Nr1d1 is over-expressed (a condition that increases cell senescence) and increases when Nr1d1 is silenced (a condition that reduces cell senescence). In contrast, in luciferase reporter assays, they show a positive regulation of Nr4a3 by Nr1d1 and of SerpinA3 by Nr4a3. Please note that NR4a3 is commonly associated to cell proliferation and survival, and it is able to increase cell life-span.
- The n of experiments should be indicated in all figure panels.
- The effect of SR8278 on Nr1d1expression was very low, and it cannot be appreciated in the western blot image used to illustrate such effect.
- Figure 1B, 1D and 3B, supplementary Figure 1A and B, supplementary Figure 2A and B. The SD value should be indicated in all bars (Ctrl or DMSO bars included).
- Results (Figure 3A). In MCM cells, Nr1d1 knockdown increased cell proliferation suggesting that the function of Nr1d1 is related to the reduction of cell proliferation. However, the authors state “Nr1d1 knockdown in MCM has affirmed the function Nr1d1 in Sca-1+CD31- cells with promoted cell proliferation”. This is absolutely confusing.
- Figure 3E. Cell photomicrographs showed to illustrate the effect of Nr1D1 knockdown on cell senescence are not representative. These images show 4 and 3 senescent cells in Ctrl and KD cultures, respectively, while bar graph indicates that this rate should be 2:1.
- Figure 5E. The same cell photomicrograph showed in Figure 3E as Ctrl is used in Figure 5E, but surprisingly the number of senescent cells indicated by arrows is different (4 and 3, respectively). As in Figure 3E, the number of senescent cells in the cell photomicrographs shown is far to be representative of the rate of senescence for both groups showed in the adjacent bar graph.
Minor points:
- English grammar should be checked. Examples: line 98, “And increased 1.25-fold increase…”; line 102, “the relevance of Nr1d1 with aged…”; line 114 (and others), “MCM has affirmed…”; etc.
- Legend of Figure 1B: is “DD-PCR” correct?
- Legend of Figure 1B. “Left lane” and “right lane” should be changed for “left or right panel” or “left or right bar graph”.
Author Response
Thank you very much for reviewing our manuscript and giving great suggestions. In the advice you gave us, we felt your profound academic attainment and deeply convinced us.
-.In the abstract, we have followed your advice and shifted the focus of the abstract from cellular remodeling to cellular senescence. “Our previous work indicated that heart derived Sca-1+CD31− cells increased Nr1d1 mRNA level with aging. However, little is known about how Nr1d1 affects Sca-1+CD31− cells senescence.”
-.In the manuscript Nr1d1 mechanistic study, the regulatory relationship between Nr1d1 and Nr4a3 and Serpina3 could not be fully clarified using Luciferase activity assay. However, in the discussion of the manuscript, we cited "Effect of ApoA4 on SERPINA3 mediated by nuclear receptors NR4A1 and NR1D1 in hepatocytes" (Zhang et al. 2017). In this paper, they performed ChIP, Luciferase activity assay and RNA interference-mediated NR4A1 or NR1D1 gene knockdown of Nr1d1 and NR4A1 to validate the relationship between ApoA4 and SERPINA3 in hepatocytes . In addition, validation of the interaction with Nr1d1 in cardiac cells with NR4A1 and Serpina3 is an important part of a Guangxi Natural Science Foundation study that we have just been approved. We will present the obtained results in a future manuscript publication.
-We have attached the N values to the figure panels and figure panels legend.
-The effect of SR8278 on Nr1d1 expression was very small, but due to the use of GSK4112 an agonist of NR1D1, to ensure the integrity of the manuscript content, we still used SR8278 to treat MCM cells. Surprisingly, the proliferation, apoptosis, and cellular senescence of SR8278-treated MCM cells yielded more desirable results. We therefore retained this part of the experiment in the supplementary material.
-In the data of Figure 1B, all data of the control group are "1" and the experimental group is the relative value corresponding to the control group, so the data of the control(or DMSO) group is 1.00+0.00. Therefore all error bars are not shown in the bar graph. The same to Figures 1D and 3B, Supplementary Figures 1A and B, and Supplementary Figures 2A and B。
-Sca-1+CD31- cells and MCM are both cardiac-derived cells. the proliferation mechanism of MCM cells may indirectly respond to Sca-1+CD31- cells. However, the use of the word "affirmed" is not very accurate, your advice is very scientific. Therefore, we have applied your sentence. “Nr1d1 knockdown increased cell proliferation suggesting that the function of Nr1d1 is related to the reduction of cell proliferation.”
-In the figure3E and 5E, there are many more stained cells that are not marked with arrows. We have marked more stained cells with arrows in the figure, making the ratio of the number of cells calibrated to 2:1 or 2:1.
-We have performed a check of the English grammar of the manuscript. If needed, we will submit the manuscript to the editor's designation for revision after the article is accepted.
-We did apply digital-PCR to perform the experiments in the early stages of the experiment. Because the number of Sca-1+CD31- cells was too few to perform a conventional qRT-PCR experiment. Because of the inconvenience of using the instrument and expensive materials for digital-PCR, and we purchased a high-speed flow cytometer with a 70um nozzle to obtain more Sca-1+CD31-cells. So for the later related experiments, we still performed with qRT-PCR.
-We have corrected all figure legend.
Thank you again for your review of our manuscript.
Best wishes!
Round 2
Reviewer 1 Report
Based on the modification the quality of manuscript massively improved.